# Experimental Investigation on the DPF High-Temperature Filtration Performance under Different Particle Loadings and Particle Deposition Distributions

**Yong Tong** [1,2], **Jie Tan** [1,2], **Zhongwei Meng** [1,2,*], **Zhao Chen** [1,2] **and Liuwen Tan** [1,2]

1 Key Laboratory of Fluid and Power Machinery, Ministry of Education, School of Energy and Power Engineering, Xihua University, Chengdu 610039, China; ty@mail.xhu.edu.cn (Y.T.); jiet@stu.xhu.edu.cn (J.T.); chenz@stu.xhu.edu.cn (Z.C.); liuwen@stu.xhu.edu.cn (L.T.)
2 Vehicle Measurement, Control and Safety Key Laboratory of Sichuan Province, School of Automobile and Transportation, Xihua University, Chengdu 610039, China
* Correspondence: mengzw@mail.xhu.edu.cn

**Abstract:** Based on DPF filtration and regeneration bench, the solid particle emission and high-temperature filtration characteristics of different carbon black particle loadings and particle deposition distributions are studied. The aerosol generator (PAlAS RGB 1000) is used to introduce carbon black particles into the inlet of a DPF, and the NanoMet3 particle meter is used to measure the solid particle concentration at the inlet and outlet of a DPF to obtain the filtration characteristics. Previous studies found that without inlet carbon black particles, there was an obvious solid particle emission peak at the outlet of the deposited DPF during the heating, and the concentration increased by 1–2 orders of magnitude. In this paper, the high-temperature filtration characteristics under steady-state temperature conditions are studied. It is found that a DPF can reduce the range of inlet fluctuating particles, and with the increase of temperature, the proportion of large solid particles in the outlet particles increases, and the size distribution range decreases. Particle loading has positive and negative effects on the DPF filtration, and the DPF has the optimal particle loading, which makes the comprehensive filtration efficiency improve the highest. The deposition transition section can make the deposition particles in the DPF uniform, but the filtration efficiency is reduced.

**Keywords:** diesel particulate filter; high-temperature filtration; particle loading; deposition distribution

## 1. Introduction

In order to reduce the emission of particulate matter from vehicle exhaust, the diesel vehicle is equipped with a wall-flow diesel particulate filter (DPF) [1] to meet the limit of solid particulate (>23 nm in diameter) number (SPN23) emission ($6 \times 10^{11}$ #/km) as early as 2011 in Europe (Euro 5b emission regulation) [2]. In addition, the same SPN23 emission limit was applied to gasoline direct injection (GDI) vehicles in 2017, and the more stringent emission standard is likely to be satisfied only by installing a gasoline particulate filter (GPF) with a similar structure to a DPF [3]. In China, DPFs were installed on new heavy-duty diesel vehicles as early as 2016 (Beijing, China 5 emission regulation). After the implementation of the China 6 emission regulation, DPF becomes the mainstream after-treatment technology for new diesel vehicles and GDI vehicles, which can effectively realize the SPN reduction, including solid particle emissions during engine start-up [4]. A DPF has a wall-flow honeycomb channel structure which can effectively reduce particle emission and meet the requirements of emission regulations [5,6], whereas the traditional DPF technology is accompanied by some defects during the regeneration process, such as additional energy input, higher regeneration temperature to melt the filter [7,8] and so on, so that the catalysts become the significant tool to protect the DPF against the spike in local temperature. This type of DPF is called a catalytic DPF (CDPF), which can not only reduce

the PM emission with high efficiency for heavy-duty diesel engines [9] but also efficiently lower regeneration temperature to protect the filter. By coating a thin catalyst layer on the DPF porous substate, the obstacle of pressure-drop of CDPF higher than the bare DPF can be solved, and the filtration property is further maintained [10]. Simultaneously, a high dispersion with deep penetration of catalyst into the filter walls is needed to enhance the contact between catalyst and soot particles to facilitate regeneration [7,10]. During the working process of the particulate filter, there is still a sudden increase of particulate emission at its outlet, mainly in the following two cases: (1) the active regeneration process of the particulate filter. It is found that compared with the absence of regeneration condition, the solid particulate number (SPN) at the DPF outlet can be increased by 3~4 orders of magnitude during the active regeneration process [11,12], and the SPN value can reach $40 \times 10^{11}$ #/km, far exceeding the SPN limit of Euro 5B ($6 \times 10^{11}$ #/km) [2]. (2) The transient conditions of the engine (starting and accelerating). Some studies have found that the SPN value of heavy diesel vehicles within 900 s after start-up can reach $400 \times 10^{11}$ #/km in the Worldwide Harmonized Vehicle Cycle (WHVC) test, far exceeding the Real Driving Emissions (RDE) SPN limit ($9 \times 10^{11}$ #/km) [13]. In addition, the sudden acceleration of the engine will also lead to a significant increase in the DPF's outlet particle number [11,14]. Acceleration under full load condition, the particle emission can reach more than 50 times the average value of the New European Driving Cycle (NEDC) test [11].

A lot of research work has been carried out to settle the problem of the sudden increase of particulate emission during the active regeneration process [15–20]. Bergmann et al. [11] and Rothe et al. [21] indicated that the new or secondary particles could be produced during the process of soot layer oxidation, which might penetrate the DPF's substrate and decrease the filtration efficiency. During the DPF regeneration process, the original exhaust particle concentration sharply increases, resulting from the sudden increase of particulate emission in transient conditions. Even if the filter efficiency of the particle filter remains unchanged, there will be dramatic growth in the particle concentration at the filter outlet [22–26]. At the same time, the transient conditions lead to the change of particle size in the inlet flow, resulting in the low filtration efficiency [23,24,27,28]. Besides, the sudden increase of inlet flow rate (filtration velocity) blows the loose particles in the filter out. During the engine cold-start period, the filter is at room temperature, and the exhaust flow temperature is low, which leads to the decrease of particle filtration efficiency [29].

The above-mentioned researches have been carried out on the phenomenon of abrupt increase of particles at the outlet of the DPF under active regeneration and transient conditions, indicating that filtration efficiency is tied to the operation condition of the DPF, thus affecting the emission of particulates. Consequently, in order to further control and reduce particulate emission, it is urgent to deeply investigate the filtration characteristics of a DPF under various operating conditions. Focusing on the DPF filtration performance, Bergmann et al. [11] measured the particle emission at the outlet of a light-duty vehicle with a DPF, and reported that the DPF had a high capture efficiency for particles of various sizes, the peak of particle number concentration at the DPF outlet was about 60 nm. Based on the heavy-duty diesel engine bench, Shuai et al. [30] found that the outlet was mainly composed of aggregated particles, and the peak particle concentration was also distributed at about 60 nm. Lou and Tan et al. [31] indicated that the particle size distribution at the CDPF outlet presented multi-peak shape, and the peak particle size was about 10 nm, 20 nm and 60 nm, respectively, and the particles with the size of 7–15 nm had strong penetrability [32]. Numerical simulation showed that the filtration efficiency of bare DPF substrate was only 40–50%, while the substrate with a soot loading of 0.2 g could improve the filtration efficiency to more than 90% [33]. The filtration efficiency of the newly regenerated DPF substrate for 10–40 nm particles could be as high as 99%, while for 100 nm particles, the filtration efficiency rapidly dropped to less than 80% [34]. It follows that there are some differences in the particle size distribution at the outlet of the DPF with the variation of fuel characteristics, operating conditions and other factors of the engine, and different measurement methods also enable particle penetration characteristics to be diverse.

In fact, particle penetration characteristics linked to the filtration efficiency are closely related to particle size and loading condition of the DPF. However, the influencing mechanism of the various operating parameters of the DPF on particle penetration characteristics is not be fully revealed in the existing investigations. Therefore, the effects of filtration temperature, particle loading and particle deposition distribution on particle penetration characteristics deserve profound exploration. In this work, the filtration performances of the DPF under different filtration temperatures, particle loadings and particle deposition distributions are analyzed based on the DPF filtration and regeneration test bench aiming to further investigate the DPF filtration characteristics and lay an experimental foundation for particle emission reduction at the outlet of the DPF.

## 2. Materials and Methods

The DPF used for this bench is purchased locally, and it was made of cordierite without any catalytic coating. The size of the DPF is 144 mm × 152 mm (diameter × length), and numbers of channels are 100 cells per square inch (CPSI), with the porosity about 45% and mean pore size about 12 μm. The components of soot were affected by engine size, fuel and operating conditions, so that carbon black (Printex-U, PU) was widely used as diesel surrogate commercial soot [35–38]. In this work, carbon black is supplied by Degussa, and the measured carbon black particle number is referred to SPN. The physical properties of carbon black are summarized in Table 1.

**Table 1.** Physical properties of carbon black.

| Diameter (nm) | BET (m²·g⁻¹) | Volatile (%) | Oil Absorption (g·(100 g)⁻¹) | Ash Content (%) |
|---|---|---|---|---|
| 25 | 92 | 5 | 460 | 0.02 |

Figure 1 shows the carbon black loading system, which is described in our previous papers [18,19,39,40]. The pre-filter and air dryer are used to remove water drop and impurities in the air, respectively. The airflow rate is controlled by a pressure regulating valve before flowing into the particle generator. The carbon black particles in the particle generator are raised to form a stable particle aerosol and then flowed into the DPF channels. Inlet transition section with length of 50 cm is applied to obtain uniform velocity and particle concentration inlet for the DPF, which has been described in our previous paper [18].

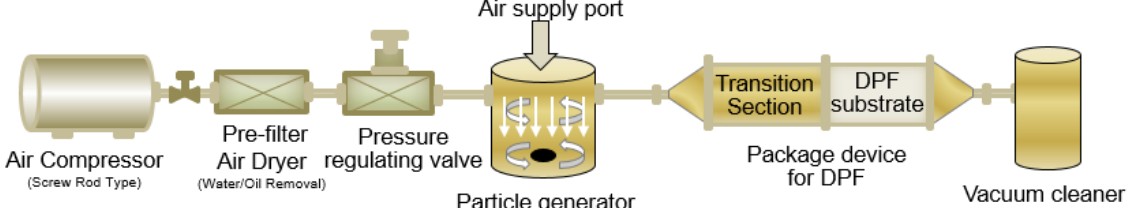

**Figure 1.** Schematic of the DPF loading system [18].

Figure 2 shows the schematic of the DPF filtration and regeneration test bench with additional heating sources. In our previous experimental work, this bench is mainly used to test the DPF regeneration performance [18,19,39,40], while in this paper, this bench is mainly used to test the DPF filtration performance. The pre-filtered air from air compressor flowed through a mass flow meter, the electrical heater (LE10,000 DF HT, LEISTER) and the DPF substrate, respectively. The air is heated to the setting temperatures by the electrical heater, and the temperatures in the DPF are recorded by LabVIEW software. A thermal insulation cloth is used to reduce heat loss during the filtration and regeneration process. At the front of the electrical heater, a particle generator (PALAS RGB 1000) is used to bypass the carbon black particles into the main inlet flow. The carbon black particle aerosol is further dispersed and mixed in the heater and finally enters the DPF substrate for filtration.

The particle emission characteristics are measured by NanoMet3 (Matter Aerosol AG, Wohlen, Switzerland). During the test, the measurement position of NanoMet3 is always at the outlet of the DPF substrate, and the particle concentration at the inlet and outlet of substrate can be obtained by installing the DPF substrate or not. The filtration efficiency can be obtained by comparing the particle concentration at the inlet and outlet. The pressure differences between the inlet and the outlet of the DPF are measured by a pressure sensor. The particle loading capacity of the DPF, the mass flow rate of incoming air, the volume fraction of incoming oxygen (compressed air) and the temperature range are 0–10 g/L, 16.8 g/s, 21%, 25–400 °C, respectively.

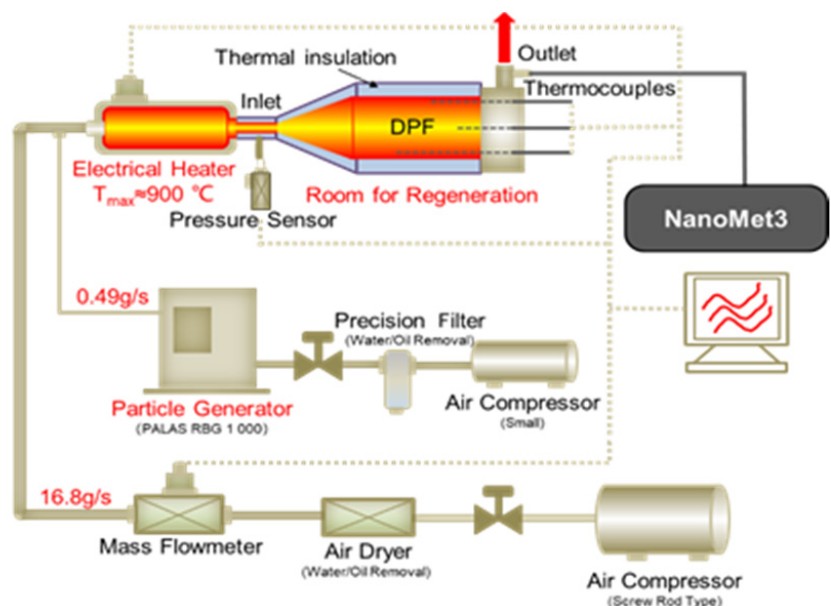

**Figure 2.** Schematic of the DPF regeneration test bench.

## 3. Results and Discussion

### 3.1. Particle Emission Characteristics in Continuous Heating Process

In our previous DPF regeneration experiment, without adding carbon black particles into the main flow, we found that there was an obvious particle emission peak at the outlet of the deposited DPF and the particle concentration increased by 1–2 orders of magnitude. Those particles are mainly due to the escape of small particles already deposited inside the DPF substrate channels under the action of high-temperature airflow. With the increase of inlet temperature, the gas viscosity increases and the small particles deposited in the micro-pores on the wall of the DPF channel, and those with small adhesion are easy to penetrate and escape under the action of gas flow [18].

As shown in Figure 3, the temperature window of the DPF outlet particle emission is between 150 °C and 450 °C, which indicates the temperature is an important factor affecting the penetration and escape of deposited particles. At the same time, high temperature can make the micro-pores in the particle layer deposited in the DPF substrate expand, resulting in the decrease of filtration efficiency [16,41]. Therefore, it is necessary to further study the steady-state filtration and particle release characteristics of the DPF substrate at various temperatures.

In addition, it can be seen from Figure 3 that the peak width of particle concentration discharged from the outlet of the DPF substrate with 50 cm transition section length ($L_T$ = 50 cm) is significantly larger than that of the DPF substrate without transition section ($L_T$ = 0 cm). The particles deposited in channels of the former DPF are more uniform than the latter [18], which indicates that different particle deposition distribution in the DPF will lead to different particle penetration and emission characteristics. Therefore, it is necessary to further explore the influence of different particle deposition distribution on particle filtration and particle emission characteristics.

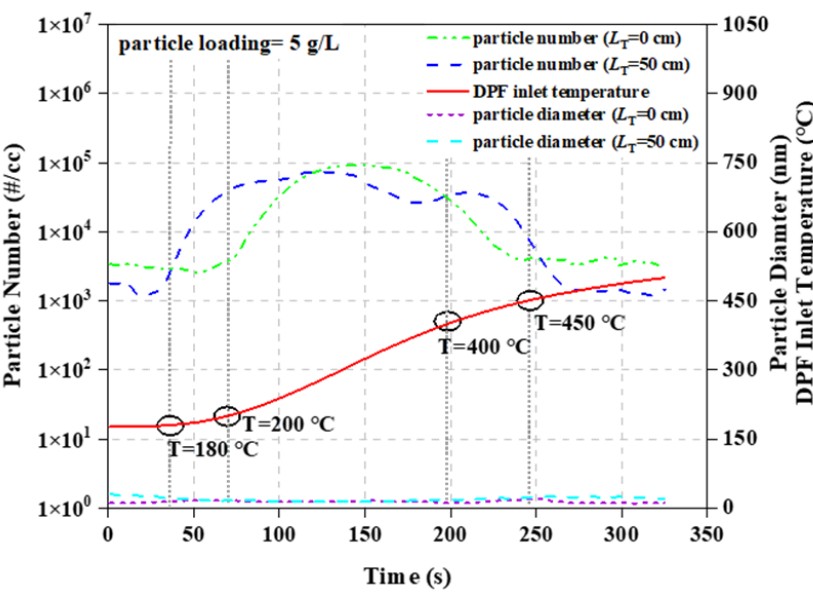

**Figure 3.** Particle emission characteristics with different inlet transition section lengths during heating.

### 3.2. Filtration Characteristics at Constant Temperature

As mentioned above, the temperature window of the DPF outlet particle release is not more than 450 °C. However, due to the time lag of temperature rise inside the DPF substrate, 400 °C is selected as the highest temperature to carry out a steady-state filtration experiment. The PAlAS RGB 1000 particle generator is used to form a high concentration particle stream at the inlet of the DPF. The particle concentration at the inlet of the DPF substrate is maintained between $1.0 \times 10^4$ (#/cc) and $1.0 \times 10^6$ (#/cc), and the particle size is between 50 nm and 300 nm, which are shown in Figure 4. The fluctuation of particle concentration and particle size is mainly due to the characteristics of the PALAS RGB 1000 particle generator and the PU carbon black. Although PU carbon black is commonly used to simulate diesel particulate matter [35–38], there is a certain degree of agglomeration of PU particles. When the brush of the particle generator rotates, it causes fluctuations of particle concentration and particle size, which is different from the real steady-state of concentration and particle size of exhaust particles in the diesel engine filtration process. However, this paper only aims to investigate the filtration characteristics of the DPF. By comparing the particle concentration at the inlet and outlet, the filtration efficiency and the change characteristics of particle size can be obtained.

It can be seen from Figure 4 that in each steady-state temperature filtration experiment, the inlet particle concentration and particle size characteristics have good repeatability within 500 s for different tests, which ensures the reliability of the test data of the experimental system. For the particle concentration and particle size at the DPF outlet, the fluctuation range and frequency of the particle number concentration and particle size at the outlet decrease with the increase of the filtration temperature. This trend is especially distinct when the filtration temperature is 400 °C, which indicates that the DPF has a good reduction effect on the fluctuation of the particle at the inlet. Simultaneously, the number concentration of particles at the outlet is significantly lower than that at the inlet, which is mainly due to the filtration and interception effect of the DPF on inlet particles. It must be pointed out that the particle number concentration test method used in this paper is: measure the particle concentration without the DPF substrate in the first, and then install the DPF substrate after the inlet particle concentration is stable, and test the particle concentration at the outlet, so the particle concentration at the inlet and outlet is obtained in different time periods. Therefore, the filtration performance of the DPF substrate cannot be directly obtained when the inlet particle concentration changes transiently. The filtration efficiency can only be obtained by comparing the total inlet and outlet particle concentration to evaluate the filtration performance of the DPF substrate.

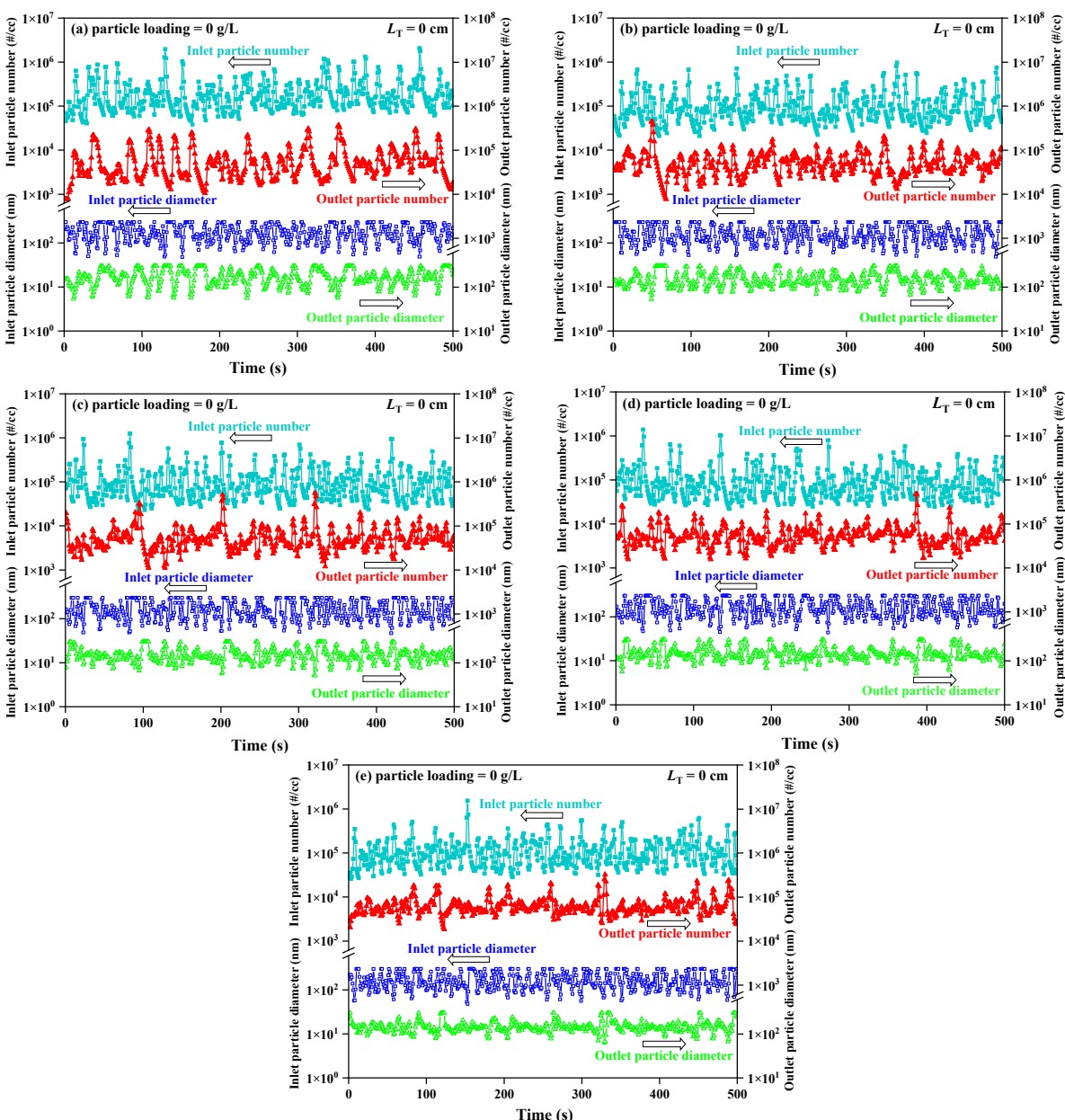

**Figure 4.** Particle concentration and particle size distribution of DPF inlet and outlet in steady filtration experiment with particle loading of 0 g/L, transition section length $L_T$ of 0 cm under different filtration temperatures: (**a**) 25 °C; (**b**) 100 °C; (**c**) 200 °C; (**d**) 300 °C; (**e**) 400 °C.

It is not easy to obtain an intuitive comparing the particle size changes of the DPF carrier at the inlet and outlet through Figure 4. Therefore, this paper summarizes and counts all the particles at the inlet and outlet during the 500 s test time. The particle size and concentration distribution are shown in Figure 5a,e and the statistical probability results of particle size distribution are shown in Figure 5f. During the test time, the average particle size distribution probability of particles has a certain distribution. By comparing the average particle size distribution probability of inlet and outlet particles, the filtration performance of the DPF can be indirectly evaluated and measured. It can be seen from Figure 5a–e, the number and concentration of particles at the inlet and outlet of the DPF decrease with the increase of particle size, which is in line with the characteristics of aerosol particles: the number concentration of particles with small size is larger. In addition, it also can be seen that the distribution probability of outlet particles in the large particle size range is larger than that of inlet particles, and this trend is more obvious with the increase

of filtration temperature. Figure 5f shows the particle size distribution probability of inlet and outlet particles in each particle size range. With the increase of filtration temperature, the peak particle size of particle size distribution probability gradually increases, and the particle size distribution range of outlet particles gradually decreases. The reasons are as follows: on the one hand, the DPF has a higher filtration efficiency for small particles, while with the increase of particle size, the filtration efficiency gradually decreases [33], which results in an increase in the proportion of large particles at the outlet of the DPF. On the other hand, with the increase of filtration temperature, the viscosity of the airflow increases, and the possibility of the deposited particles escaping from the DPF by the airflow increases. Compared with the small particles, the adsorption capacity of the large particles adsorbed in the micro-pores of the wall of the DPF substrate is smaller, which is easy to be taken away by the high temperature and high viscosity airflow, so that it easy to cause the increase in the proportion of large particles.

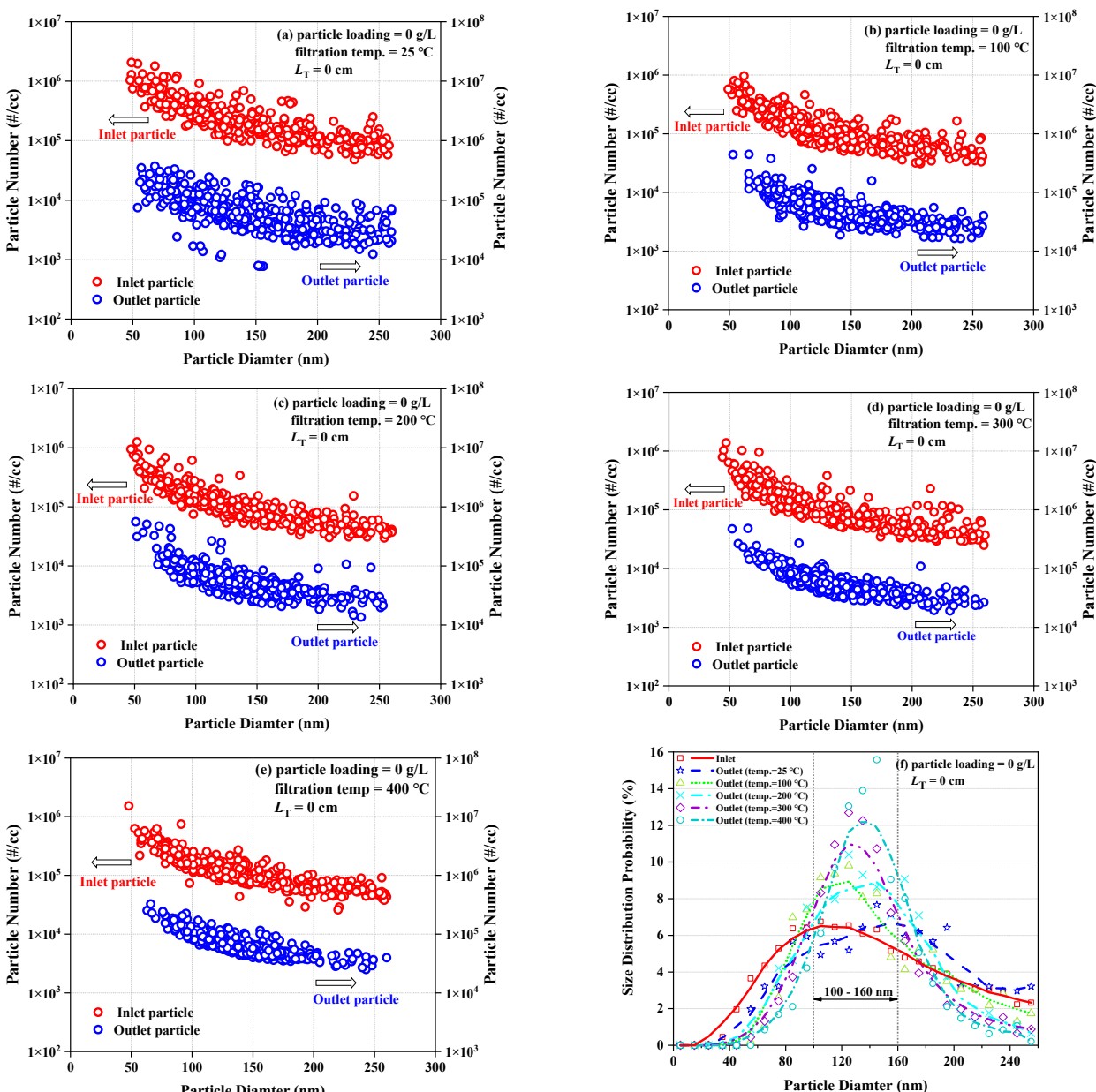

**Figure 5.** The particle number of inlet and outlet particle with particle loading of 0 g/L, transition section length $L_T$ of 0 cm under different filtration temperatures: (**a**) 25 °C; (**b**) 100 °C; (**c**) 200 °C; (**d**) 300 °C; (**e**) 400 °C; (**f**) size distribution probability of particle diameter in DPF inlet and outlet.

### 3.3. Effect of Particle Loading on Filtration Characteristics

By comparing the total number of particles at the DPF inlet and outlet, the filtration efficiency of the DPF with different particle loading is obtained, as shown in Figure 6a. It can be seen from the figure that the filtration efficiency gradually decreases with the increase of filtration temperature. This is related to the micro-pore expansion of the deposited particles caused by high temperature, which reduces the filtration efficiency [16,41]. On the other hand, the viscosity of high-temperature airflow increases with the temperature, and it is easy to take away the deposited particles, which reduces the filtration efficiency. When the particle loading of the DPF is 2.5 g/L, the filtration efficiency is the highest for different filtration temperatures. The filtration efficiency reaches about 90% at 25 °C and near 80% at 400 °C. With the increase of particle loading to 5 g/L and 10 g/L, the filtration efficiency decreased gradually. When the filtration temperature is 400 °C, the filtration efficiency of 10 g/L of the DPF is only 54.3%, which is lower than that of bare DPF substrate (0 g/L). On the one hand, with the increase of particle loading, particles deposited on the wall of the DPF channel increase. Although the thickening of the particle layer is easy to intercept inlet particles, more deposited particles are easy to be carried out of the DPF by inlet flow, resulting in the decrease of filtration efficiency. On the other hand, the relative deviation of particles deposited in each channel decreases with the increase of the loading capacity. As shown in Figure 3, compared with the DPF with non-uniform deposition, the DPF with uniform deposition of particles is easier to release particles during the heating process, resulting in the decrease of filtration efficiency. The above analysis shows that the DPF substrate has the best particle-loading value, which can not only intercept the inlet particles but also prevent the deposited particles from being carried out of the DPF by the airflow, making its filtration efficiency the highest.

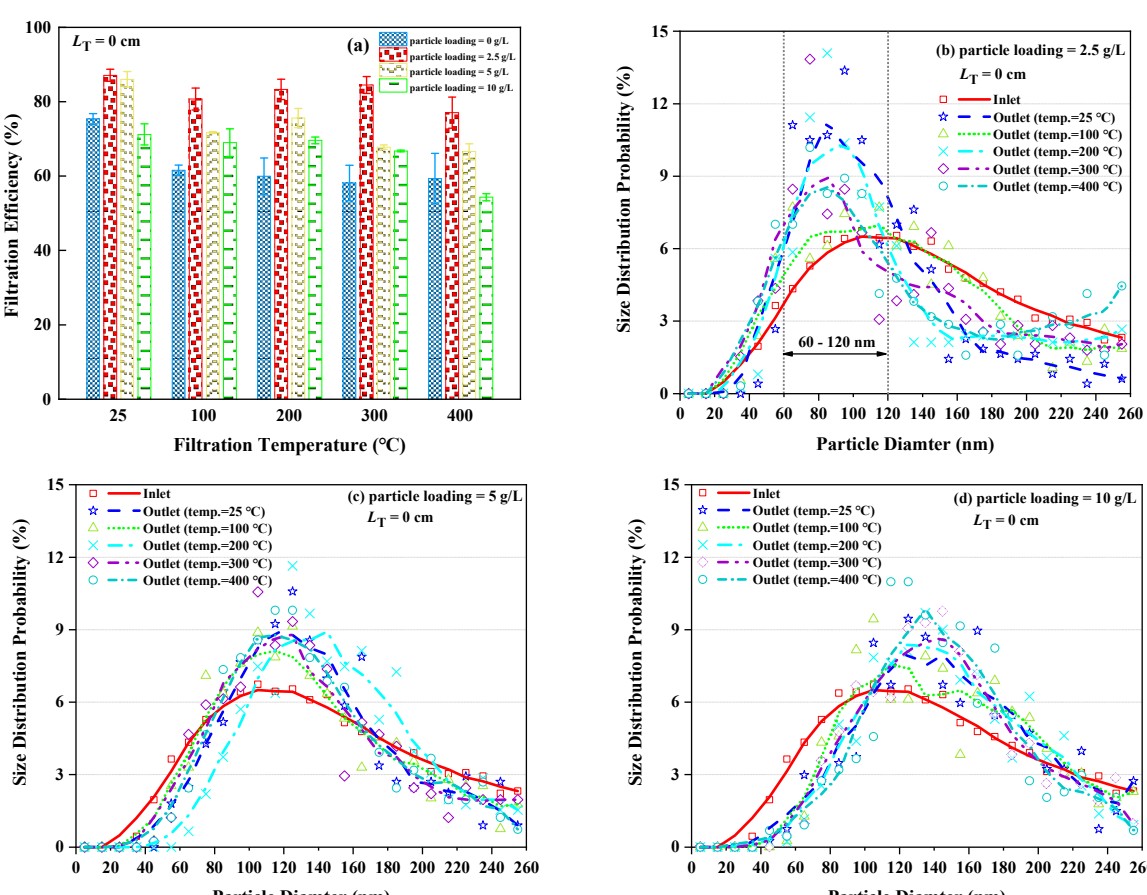

**Figure 6.** (**a**) The filtration efficiency of particle loading from 0 g/L to 10 g/L; particle size distribution probability of different temperatures in DPF inlet and outlet with transition section length $L_T$ of 0 cm under different particle loadings: (**b**) 2.5 g/L; (**c**) 5 g/L; (**d**) 10 g/L.

The probability of particle size distribution at the outlet is shown in Figure 6b–d for 2.5 g/L, 5 g/L and 10 g/L particle loading, respectively. It is worth mentioning that, different from the bare substrate (0 g/L, Figure 5f) and the particle loading of 5 g/L and 10 g/L (Figure 6c,d), when the loading capacity is 2.5 g/L, the peak value of particle size distribution at the outlet of the DPF tends to change to smaller particle size. It is indicated that when the loading capacity is 2.5 g/L, the particles carried out of the DPF by the inlet airflow are less, and the DPF also has high filtration efficiency, which makes the particle size at the outlet smaller, and the main particle emission is about 80 nm. On the other hand, Figure 6b–d show that the particle size distribution at the outlet of the DPF gradually decreases with the increase of filtration temperature. The result is related to the fact that high temperature makes the inlet particles easy to penetrate and the deposited large particles easy to be carried out of the DPF by the inlet flow.

### 3.4. Influence of Transition Section on Filtration Characteristics

As mentioned in Section 3.3, the relative deviation of particles deposited in each channel of the DPF substrate is low for 10 g/L particle loading, resulting in the growth of particles at the DPF outlet, which is consistent with the conclusion from Figure 3. Therefore, this section further discusses the condition that when the particle loading is 10 g/L, the 50 cm transition section is used to deposit particles to further reduce the relative deviation of the deposition amount of each channel and explore the particle filtration characteristics of the DPF when each channel is uniformly deposited, as shown in Figure 7. It compares the filtration efficiency of the DPF substrate with or without 50 cm transition section ($L_T$ = 50 cm) for 10 g/L particle loading. It presents that the filtration efficiency of $L_T$ = 50 cm is significantly lower than that of the case without transition section ($L_T$ = 0 cm), and the filtration efficiency under various temperatures is maintained at about 60%. Only at 400 °C, the filtration efficiency is 60.6%, which is slightly higher than that of without transition section ($L_T$ = 0 cm). The results show that at 400 °C, with the addition of 50 cm transition section ($L_T$ = 50 cm), the particles deposited in each channel are uniform, and the particles carried out by the airflow increase. However, compared with the non-uniform deposition ($L_T$ = 0 cm), the interception effect of each channel (especially the channels at the radial edge of the substrate) on the inlet particles is enhanced so that the filtration efficiency of $L_T$ = 50 cm is greater than that of $L_T$ = 0 cm.

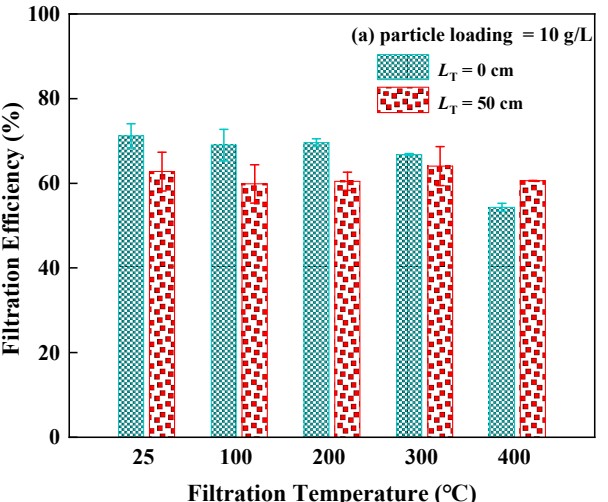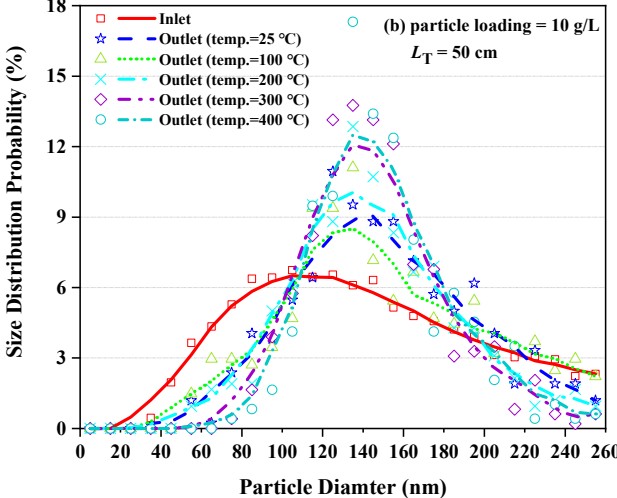

**Figure 7.** (**a**) Filtration efficiency for different transition section lengths with particle loading of 10 g/L; (**b**) size distribution probability of particle diameter in DPF inlet and outlet with particle loading of 10 g/L, transition section length $L_T$ of 50 cm.

Figure 7b shows the probability of particle size distribution at the outlet when a 50 cm transition section is added. Comparing Figures 6b and 7b, it can be found that the proportion of outlet particles in the large particle size range (110–170 nm) increases, and the particle size distribution range decreases accordingly when $L_T$ = 50 cm. Unfortunately, it is impossible to make a quantitative comparison between the mechanism of the particles in the deposition layer being blown out under the action of the inlet flow and the filtration of the inlet particles in this experiment. The follow-up experiment can rely on the single-channel filter bench [42] to explore the particle release mechanism of particle layer structure (deposition thickness and deposition distribution, etc.) under the effect of inlet high-temperature flow and the filtration mechanism of inlet particles.

### 4. Conclusions

The filtration characteristics of the DPF in the temperature range of 25–400 °C under different particle loadings and particle deposition distributions are studied utilizing the DPF filtration and regeneration bench. It can be found that compared with particles at the inlet of the DPF, the fluctuation of particle concentration and particle size range of the outlet particles decrease, which becomes more obvious with the increase of filtration temperature. With the increase of temperature, the proportion of large particles in the outlet particles increases, and the size distribution range decreases due to the smaller adsorption capacity. Filtration efficiency of the DPF for inlet particles increases with an increase of particle loading, while the deposited particles can also be blown out of the DPF by the inlet flow, resulting in the filtration efficiency decreasing so that the optimal particle loading is 2.5 g/L. The filtration efficiency of the DPF with a transition length of 50 cm is lower than that of transition length of 0 cm, while the former is higher than the latter only at 400 °C due to the enhanced interception effect of each channel in the DPF.

The results obtained in this work can provide technical support and guidance for the design and regeneration strategy of DPFs, and further optimize particle-penetration characteristics by improving filtration properties. In the future, the effects of particle layer structure under high-temperature inlet flow on particle filtration and emission mechanism will be investigated elaborately, and the influence of catalysts and ashes on the above will be taken into consideration.

**Author Contributions:** Conceptualization, Y.T. and Z.M.; Data curation, Y.T.; Formal analysis, J.T. and Z.C.; Funding acquisition, Z.M.; Investigation, L.T.; Methodology, Z.M.; Project administration, Y.T. and Z.M.; Resources, Y.T. and Z.M.; Software, J.T.; Supervision, Z.M.; Validation, Y.T. and Z.M.; Visualization, Y.T.; Writing—original draft, J.T.; Writing—review and editing, Y.T. and Z.M. All authors have read and agreed to the published version of the manuscript.

**Funding:** This research was funded by the National Natural Science Foundation of China, grant number 52076182; the Science & Technology Department of Sichuan Province, grant number 2019YFS 0499 and 2019YJ0594; the Sichuan Provincial Scientific Research Innovation Team Program, grant number 17TD0035.

**Institutional Review Board Statement:** Not applicable.

**Informed Consent Statement:** Not applicable.

**Data Availability Statement:** The data presented in this study are available on request from the corresponding author.

**Acknowledgments:** The authors are would like to thank Qian Zhang of School of Automobile and Transportation of Xihua University for her help with the reasonable discussion on topics related to this work and formatting this paper. This work has been supported by (1) the National Natural Science Foundation of China (52076182); (2) the Science & Technology Department of Sichuan Province (2019YFS 0499, 2019YJ0594); (3) the Sichuan Provincial Scientific Research Innovation Team Program (17TD0035).

**Conflicts of Interest:** The authors declare no conflict of interest.

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
