# Peer review of "Experimental Investigation on the DPF High-Temperature Filtration Performance under Different Particle Loadings and Particle Deposition Distributions"

_processes, doi:10.3390/pr9081465_

Round 1

Reviewer 1 Report

This is an interesting paper addressing an important issue. It can be considered for publication in Processes. However, I have the following comments that the authors should carefully implement in the revised manuscript before publication.

1) Introduction - The authors should explicitly state that CDPF means catalytic/catalyzed DPF. They should also highlight the importance of using suitable techniques of catalyst deposition onto the DPF from both the perspectives of filtration and regeneration. More specifically, the filtration properties of the bare (uncoated) filter must be retained as much as possible or improved upon catalyst deposition and, at the same time, a high dispersion with a deep penetration of catalyst into the filter walls is need to enhance the contact between catalyst and soot particles, thus facilitating regeneration. In this regard, the following works should be cited: Applied Catalysis B: Environmental, Volume 197, 2016, Pages 116-124; Topics in Catalysis, Volume 64, 2021, Pages 256-269.

2) Introduction - The connection between the aim of the work and the literature gaps should be better described, thus giving more strength to the reason behind this work.

3) Results and discussion/Conclusions - The practical impact of the results obtained in this work should be better highlighted (the section "Conclusions" cannot be a list of outcomes, and more criticism is needed).

4) Conclusions - The authors should also give an outlook on future research work.

I’m willing to review the revised manuscript.

Reviewer 2 Report

This paper discusses a laboratory bench using black carbon generation couple with temperature control and a DPF. The authors did not include details on the DPF and what technology engines it represents. In addition, the authors did not demonstrate the usefulness of the method on a comparative basis with that of a diesel engine application in order to demonstrate the application to that of a diesel engine or that to a gasoline engine with GPF in a broader sense.

Below are some specific comments:

Line 39, while a DPF has a high filtration efficiency, most GPFs do not have such as high efficiency like the DPFs. Please revise your statement, or provide references specific to this claim if any. GPF does not have to be as efficient as a DPF, for meeting emissions regulations.

Throughout the article, please clarify your statement about PN or call it solid PN to be specific, or to be more concise, please use SPN23 for solid particles > 23 nm in diameter. In principle, you need to state very clearly what you are measuring.

Table 1, correct the unit on the diameter to nm and not mm

It is NanoMet 3 and not Nanometer 3, please correct. Also, when was the last time this instrument was calibrated before the study.

Please provide calibration data on the instrument. This instrument uses diffusion charging to determine particle concentration and subject to potential artifact due to electrometer drift. Since you are not measuring upstream and downstream of the DPF simultaneously, how did you insure the stability of the inlet concentration. Also, how did you insure that the inlet concentration is not changing as a function of backpressure of the DPF due to DPF backpressure? Please provide data on the relationship between backpressure and inlet concentration.

Line 171. You mentioned that the inlet concentration is in the range of 1e4 to 1e6 and this is a high concentration. This is not a high concentration compared to diesel engines at the inlet of the DPF. Engine out concentration at the inlet of the DPF is significantly higher. Also, why the particle diameter in Figure 3 is much smaller than the particle diameter in Figure 4.

In Figure 3, there is a claim that during regeneration particle escape from the DPF through the outlet increasing concentration. I did not see the evidence that these particles are soot particles escaping. Also, in the case of diesel and gasoline, the particle are smaller than the one used here. So, this could be an artificat of the measurement.

Although during regeneration particles emitted at the outlet of the DPF is usually high in an actual engine, the explanation for that is much more complicated than soot particles escaping the surface of the DPF. The measurement method is an operational definition and it does not mean it is measuring soot or solid particles in the true sense of the definition. 

For Figure 5, it is not clear how the size distribution was obtained and what instrument was used for the size distribution measurement. Also, the inlet size distribution characteristics in terms of mean size, profile, geometric width do not represent that of gasoline or diesel engines. How can we then interpret these results and relate them to that of diesel and gasoline engines.

Please refrain from using the word easily in the conclusion multiple times.

Round 2

Reviewer 1 Report

The authors have addressed my comments in a satisfactory manner. Overall, the manuscript has been improved after revisions. I have only one suggestion that should be implemented in the revised manuscript before publication: the fact that, during regeneration, DPFs may undergo disruptive temperature rises leading to their melting should be corroborated by citing pertinent literature (see, e.g., Chemical Engineering Science, Volume 137, 2015, Pages 69-78).
